# The Relationship Between Socioeconomic Status and Health Behaviors in Older Adults: A Narrative Review

**DOI:** 10.3390/healthcare13141669

**Published:** 2025-07-10

**Authors:** Hidetaka Hamasaki

**Affiliations:** Japanese Academy of Health and Practice, 4-29-16 Tsukushino, Machida 194-0001, Tokyo, Japan; h-hamasaki@umin.ac.jp

**Keywords:** older adults, aging society, socioeconomic status, education, income, occupation, healthcare, digital health

## Abstract

**Background**: In rapidly aging societies like Japan, socioeconomic status (SES) plays a critical role in shaping older adults’ health behaviors. Disparities in SES influence access to healthcare, engagement in health-promoting activities, and the adoption of digital health technologies. This narrative review synthesizes current evidence on how SES affects health behaviors among older adults and highlights challenges in promoting equitable and sustainable healthcare in aging populations. **Methods**: A PubMed search was conducted for English-language articles published up to May 2025 using the keywords “socioeconomic status”, “older adults”, and terms related to health behaviors. Studies were included if they focused on individuals aged 65 or older and examined associations between SES and healthcare use, digital health, complementary and alternative medicine (CAM), supplements, or lifestyle behaviors. **Results**: A total of 24 articles were identified. Higher SES—typically measured by income, education, and occupation—was consistently associated with an increased use of preventive services, digital health tools, CAM, and healthier lifestyle behaviors such as diet, physical activity, and sleep. In contrast, lower SES was linked to healthcare underuse or overuse, digital exclusion, and less healthy behaviors. Structural and regional disparities often reinforce individual-level SES effects. Comorbidity burden and shifting health perceptions with age may also modify these associations. **Conclusions**: SES is a key determinant of health behavior in older adults. Policies should focus on redistributive support, digital inclusion, and SES-sensitive health system strategies to reduce disparities and promote healthy aging in super-aged societies.

## 1. Introduction

In aging societies, disparities in socioeconomic status (SES) among older adults have emerged as a significant public health concern. In Japan, recognized as a super-aged society, evidence indicates that lower SES among older adults is associated with adverse health outcomes, including an increased risk of cognitive decline and diminished functional capacity [1]. A longitudinal study conducted in Tama City, Tokyo, revealed that lower SES correlates with a higher need for long-term care among the urban elderly population [2,3,4]. Furthermore, research has demonstrated that income and education levels are linked to transitions in health status among community-dwelling older adults in Japan, suggesting the significant impact of socioeconomic factors on health trajectories [5]. These findings suggest that socioeconomic disparities among older adults contribute to public health challenges, necessitating the development of policies aimed at mitigating such inequities. Implementing strategies to address SES-related health disparities is imperative for constructing a sustainable society that supports healthy aging and equitable access to healthcare resources [6].

SES demonstrably shapes a wide spectrum of health-related behaviors among Japanese older adults. A cohort analysis of 17,991 community-dwelling older adults found that the odds of receiving pneumococcal vaccination were 1.76 (95% confidence intervals (CIs), 1.17–2.76) in individuals whose annual municipal tax category was not classed as “low income”, compared with low-income individuals, even after adjustment for age, comorbidity, and service access factors [7]. Digital care shows an even steeper gradient: in a nationwide web survey of 514 adults aged ≥ 60 years, telemedicine use during the COVID-19 pandemic was more than twice as common among university graduates (adjusted odds ratio (OR) = 2.63; 95% CI, 1.11–6.23) and 2.45-fold higher in senior high school graduates, relative to those whose schooling ended at the junior high level; higher family income produced a comparable effect size, underscoring the dual digital and economic divide in access to online health services [8]. Lifestyle modification also reflects SES: a population study from northern Japan that followed older adults through pandemic restrictions reported that men in the lowest income/education tertile were less likely to maintain recommended moderate-to-vigorous physical activity (MVPA) (OR = 0.49; 95% CI, 0.30–0.82), whereas women with robust social-participation networks, an SES-linked resource, had greater odds of sustaining activity (OR = 1.67; 95% CI, 1.13–2.45) [9]. Evidence for complementary and alternative medicine (CAM) also shows a similar pattern: among 200 Okinawan older adults, the prevalence of home-remedy use rose to 71.9% in women and was strongly associated with indicators of greater material and social resources, practitioners with ongoing food-control habits (often facilitated by higher income) had an adjusted OR of 4.3 for continued use, while those receiving informal caregiving support (a proxy for social capital) showed an OR of 4.2 [10]. Collectively, these quantitatively consistent relationships indicate that SES not only governs the uptake of preventive clinical services but also modulates engagement with emerging digital healthcare and self-directed health practices.

Outside Japan, systematic reviews have also reported that a low SES is significantly associated with poorer outcomes in emergency care [11], excessive gestational weight gain [12], and higher incidence of frailty [13]. However, there are very few reviews that specifically examine the relationship between SES and health behaviors among older adults.

Therefore, investigating the relationship between SES and health behaviors among older adults and identifying the associated challenges is of critical importance for the development of sustainable healthcare systems in the future. This narrative review aims to synthesize findings from previous studies and explore the conditions necessary for achieving sustainable and high-quality healthcare in an aging society.

## 2. Methods

This review provides a narrative summary to highlight the relationship between SES and various health-related behaviors in older adults, aiming to address public health issues arising from SES inequities. This review follows a structured literature review methodology proposed by Turnbull et al. [14] to ensure the quality of the review: (1) research question; (2) justification; (3) literature sources; (4) search parameters; (5) data cleaning; (6) information synthesis.

### 2.1. Research Question

The following question was formulated to guide this review:

Generally, individuals with a higher SES have greater health awareness and literacy and are more likely to engage in healthier behaviors; however, is this also true for older adults?

### 2.2. Literature Sources

The author performed an electronic literature search on PubMed for articles published up to May 2025.

### 2.3. Search Parameters

The search terms used were “socioeconomic status”; “health behavior”, “digital health”, “telemedicine”, “complementary therapies”, “diet”, “physical activity”, and “sleep”—all of which are included as Medical Subject Headings (MeSH); and “older adults”. All types of articles were considered from the database’s inception (*n* = 6880). Of these, clinical studies, including cross-sectional and observational studies, were deemed eligible (*n* = 1098). The titles and abstracts of the identified articles were reviewed to assess their relevance. The author also reviewed the reference lists of the selected articles to identify further eligible studies.

### 2.4. Data Cleaning

The article selection criteria included the following: (a) written in English; (b) involving older adults aged 65 years or older; (c) including objectively evaluated SES variables (e.g., education, income, and occupation); and (d) addressing health behaviors such as healthcare utilization, digital health, CAM, supplements, diet, physical activity (PA), and sleep. The exclusion criteria were as follows: (a) article types such as reviews, case reports, editorials, commentaries, letters, and protocols; (b) studies that did not include SES variables; (c) studies without quantitative analysis; and (d) preprints.

A total of 24 studies were included in this review.

## 3. Results

### 3.1. Healthcare Utilization

Many studies have examined how differences in SES shape healthcare utilization, including clinic visits, hospital admissions, preventive services, pharmacotherapy, and rehabilitation, among older adults. A recent systematic review and meta-analysis of 54 observational studies found that older individuals with the lowest educational attainment had 21% higher odds of receiving polypharmacy (≥5 concurrent drugs) than their most-educated peers, with a concordant disadvantage for low income, wealth, occupation, and social class, illustrating a pervasive SES gradient in pharmacotherapy [15]. Complementing these medication-focused findings, a narrative synthesis of 20 cross-sectional studies from low- and middle-income countries showed a consistent “pro-rich” pattern in primary-healthcare utilization: higher income, higher education, formal-sector employment, and enrolment in contributory insurance were each associated with more frequent outpatient contact, whereas uninsured or economically inactive older adults were more likely to forgo care [16]. Notably, this review also documented context-specific exceptions, such as free primary-care programs in Cuba, Brazil, Thailand, and parts of China, where healthcare utilization shifted towards poorer groups, indicating the potential for policy to mitigate entrenched inequities [16]. Evidence from Asia mirrors these global trends. An analysis of 10,922 participants in the China Health and Retirement Longitudinal Study (2011–2018) showed that, relative to the poorest quintile, the wealthiest quintile had 1.4-fold higher odds of any outpatient visit and almost double the rate of hospital admissions, even after adjustment for multimorbidity. Moreover, chronic disease increased service use overall but did not attenuate these SES gaps, suggesting that financial capacity rather than clinical need drives differential access [17]. These findings demonstrate a remarkably consistent pattern: whether the outcome is potentially inappropriate polypharmacy, routine primary-care contact, or costly inpatient care, a higher SES confers better access, while a lower SES increases exposure to both over- and underuse risks. In Asia, particularly in Japan and China, rapid population aging adds urgency to closing these gaps.

The author highlights several additional landmark studies that provide further insight into the mechanisms behind these inequalities and the policy levers available to mitigate them.

Hoeck et al. [18] examined whether equivalized household income, highest household educational level, and housing tenure predicted ambulatory care during the preceding two months using data on 4494 community-dwelling adults aged ≥ 65 years. Overall, 71% had consulted a general practitioner (GP) and 23% consulted a specialist. In models adjusted only for age and sex, older adults with a monthly income of EUR 750–1000 were more than twice as likely to have at least one GP contact as their wealthiest peers (OR = 2.16; 95% CI, 1.19–3.93), and those without any formal educational qualifications were more likely to visit a GP than those with qualifications (OR = 1.77; 95% CI, 1.12–2.80). However, poor self-rated health itself doubled the likelihood of GP visits (OR = 2.09; 95% CI, 1.48–2.94), and after adjusting for need factors such as functional limitations, multimorbidity, residential region, and living arrangements, all socioeconomic gradients in the probability of contact with a GP or specialist disappeared. Among the older population, there were no longer significant differences in the likelihood of having at least one contact with a GP or specialist within the previous two months. This contrasts with younger individuals, among whom those with a higher SES are more likely to consult a specialist. However, disparities remain in the frequency of GP visits, with individuals with a lower SES making more visits in both the older and younger age groups.

Older adults are often physically frail, have multiple comorbidities, and experience a decline in activities of daily living (ADL). Consequently, they inevitably require more frequent medical care than younger and healthier individuals. However, SES may influence the patterns of healthcare utilization and disease outcomes. Based on the study by Mai et al. [19], SES significantly influenced healthcare access among older adults with limited ADL in China. Specifically, individuals with “rich” economic status had over seven times greater odds of accessing care compared to those with “poor” status (OR = 7.23), and those who could afford daily living expenses were more than twice as likely to access services (OR = 2.33). Regional disparities were also evident, with residents in eastern China being nearly three times more likely to access care than those in the western region (OR = 2.91). The authors concluded that financial and regional enabling factors, rather than health needs or demographic characteristics, play a decisive role in healthcare utilization among this population. These findings indicate the importance of equitable economic policies and healthcare infrastructure distribution to improve access.

Regional disparities in healthcare resources are also a pressing issue in Japan. A recent scoping review by Kaneko et al. [20] found that rural residents in Japan experience significantly poorer access to care and lower service quality, including fewer physicians and specialists, with structural inequities evident across all three dimensions of Donabedian’s model: structure, process, and outcome. A nationwide cross-sectional study examined regional disparities in home healthcare utilization among older adults in Japan [21]. The results showed substantial regional variation in both home-visit medical care and nursing care utilization across secondary medical areas. Multivariable analysis showed that a greater availability of home-visit nursing stations and home-care support clinics and a higher proportion of single-person households among older adults were significantly associated with increased service use. These findings suggest that both healthcare infrastructure and social context play key roles in determining access to home healthcare.

SES shows a parallel regional distribution, as areas with limited healthcare resources are often those with a lower average income, educational attainment, and population density. This geographic overlap between SES and healthcare infrastructure suggests that correcting the regional maldistribution of medical resources may be critical to resolving SES-based inequities in healthcare access and outcomes.

A retrospective cohort study in the Netherlands examined the association between SES and outcomes among older adults presenting to the emergency department [22]. Patients with a low SES were more frequently hospitalized than those with a high SES (62.3% vs. 52.3%), with a longer median hospital stay (6.0 vs. 5.0 days). After adjustment, low SES was not independently associated with admission overall (OR = 1.3; 95% CI, 0.9–1.4); however, among community-dwelling individuals, low SES significantly increased admission risk (OR = 1.3; 95% CI, 1.1–1.7). In-hospital mortality was initially higher in the low-SES group (5.4% vs. 3.5%); however, the association lost significance after adjustment. These findings suggest that SES impacts hospitalization risk, particularly among community-dwelling older adults.

Leff et al. [23] conducted a cross-sectional study of 6,664,124 Medicare Advantage beneficiaries aged ≥ 65 years to assess the prevalence and characteristics of home-based medical care users. Home-based medical care was defined as care delivered by physicians, nurse practitioners, or physician assistants in the patient’s home or domiciliary setting, identified through outpatient claims with a home place-of-service code. The authors found that 5.5% received home-based medical care, with users disproportionately more likely to be dually eligible for Medicaid (45.5% vs. 20.1%) and to live in socioeconomically disadvantaged areas. Notably, 70.6% of the users resided in the most deprived quintile of the Area Deprivation Index. These results suggest that a lower SES is strongly associated with home-based medical care use, indicating its role in supporting vulnerable older adults with limited access to care. This study provides important insights. In super-aged societies such as Japan, rising social security expenditures are placing increasing strain on national healthcare financing. In response, the government has implemented policies aimed at reducing long-term hospitalizations and promoting the diffusion of home-based medical care. For example, the early initiation of physician-led home-visit care in Japan can reduce end-of-life medical costs compared to hospital care [24]. This model supports patient dignity and aligns with national efforts to promote community-based integrated care amid rapid population aging. However, if individuals with a lower SES are more likely to accept home-based care, while those with a higher SES prefer specialist inpatient treatment even at greater personal cost, such divergence may lead to widening disparities in healthy life expectancy and quality of life. This would fundamentally contradict the core principle of equity in access to high-quality healthcare.

The 2008 Community Tracking Study [25] demonstrated a substantial increase in health information-seeking among U.S. adults, rising from 38% in 2001 to 56% in 2007. Internet use as a primary source doubled, from 16% to 32%. Socioeconomic disparities were prominent: 72% of adults with a postgraduate education searched for health information, compared to only 42% of those without a high school diploma; online search rates were even more skewed (52% vs. 10%). Although older adults (≥65 years) tripled their Internet use during the period, only 18% searched online compared to 36% of those aged 18–49. Among information seekers, 52% discussed findings with a clinician and 51% changed their health behavior, indicating broad downstream effects. The authors suggest that this surge reflects rising out-of-pocket costs and reduced access to physicians. However, the digital divide may exacerbate SES-related disparities in healthcare utilization by limiting access to decision-support resources among socioeconomically disadvantaged older adults. Addressing this requires targeted age-appropriate digital tools and stronger primary care integration. The generational digital divide poses a significant challenge in aging societies, and over the past two decades, various studies have explored healthcare information technology (IT) utilization among older adults. Using data from the 2009 U.S. National Health Interview Survey, Choi et al. [26] examined the relationship between health service use and health IT use among 5294 community-dwelling adults aged 65 and older. The study found that only 32.2% of those aged 65–74, 14.5% of those aged 75–84, and 4.9% of those 85 and older had used the Internet for health-related purposes in the prior 12 months. Despite low overall rates, older adults who had visited general practitioners, medical specialists, eye doctors, physical/occupational therapists, or mental health professionals were significantly more likely to use health IT than those who had not. Conversely, no significant association was found between health IT use and having undergone hospitalization, surgery, home care, or chiropractic services. Importantly, SES exerted a strong influence on health IT use. Lower educational attainment and lower income-to-needs ratios were both independently associated with significantly reduced odds of health IT engagement. For instance, individuals without a high school diploma were 88% less likely to use health IT than those with graduate degrees. Racial and ethnic minorities, particularly Hispanic and non-Hispanic Black adults, also had significantly lower odds of health IT use compared to non-Hispanic Whites. The authors concluded that while the use of general healthcare services may stimulate health IT engagement, deep SES-related disparities remain. Gender-specific patterns were also observed: older women who used specialized services were more likely to use health IT, while older men showed a unique positive association between mental health service use and health IT. More than a decade has passed since this study was conducted, and since then, a growing number of older adults have come to routinely use health-related technologies such as smartphones and wearable devices. However, the generational digital divide persists, with significant disparities in digital health engagement still observed among older adults, particularly those with a lower SES, limited health literacy, or sensory and cognitive impairments [27]. As we enter an era increasingly defined by artificial intelligence in healthcare, addressing this digital inequity remains a critical challenge for ensuring equitable access and participation in emerging health systems.

These findings suggest that health IT use among older adults reflects not only healthcare needs but also broader SES advantages. Bridging the digital divide in aging societies will require targeted interventions, such as improving digital literacy, tailoring web content to older users, and addressing infrastructural barriers, particularly for disadvantaged populations. Without such efforts, existing SES-based inequities in healthcare utilization may widen in the digital era.

Table 1 summarizes the association between SES and healthcare utilization among older adults.

### 3.2. Digital Health

The COVID-19 pandemic significantly transformed our daily lives. In healthcare, it catalyzed the rapid advancement of digital technologies, such as telemedicine, smartphone applications, electronic health records, AI and data analytics tools, thereby reshaping the future landscape of medical care. Indeed, comprehensive reviews have demonstrated that during the pandemic, telemedicine and digital health interventions saw widespread adoption globally [28]. However, it was also found that socioeconomic disparities among older adults—the digital divide—remain a significant barrier to healthcare access in this population.

Fealy et al. [29] reported that among over 21,000 Australians aged ≥ 65 years surveyed between 2020 and 2022, high telehealth use was significantly associated with being female (OR = 1.55), aged < 85 years (OR for ≥85 years = 0.61), cohabiting (OR for living alone = 0.72), living in major cities (OR for outer regional/remote areas = 0.40), and residing in socioeconomically disadvantaged areas. Health-related factors were strong predictors: individuals with disability (OR = 3.06), chronic disease (OR = 1.72), or multimorbidity (OR = 3.63), as well as those reporting poor quality of life (QOL) (OR = 1.36) or psychological distress (OR = 1.07), were more likely to engage in telehealth. The prior use of telehealth early in the pandemic was significantly associated with continued use in 2022, though no interaction was found with chronic disease or multimorbidity. These findings underscore the importance of tailoring telehealth services to older adults with complex health needs, while addressing geographic and social barriers that may hinder equitable access.

Gordon and Hornbrook [30] reported that SES plays a critical role in shaping access to and the utilization of eHealth technologies among older adults aged 65–79 years. Their analysis demonstrated marked disparities based on educational attainment and household income. Specifically, over 65% of college graduates were registered for patient portals, compared with only 36.6% of those without a high school diploma. Likewise, approximately 70% of individuals in the highest income bracket reported using the Internet for health-related purposes, in contrast to 33.9% in the lowest income group. These disparities were further exacerbated among racial and ethnic minorities, including Black, Hispanic, and Filipino individuals, and among adults aged ≥ 75 years, who exhibited significantly lower rates of device ownership, Internet experience, and willingness to engage with digital health tools. The findings emphasize SES as a fundamental determinant of digital health equity and highlight the urgent need for inclusive strategies to ensure that innovations in telehealth do not inadvertently deepen existing healthcare disparities among socioeconomically disadvantaged older populations.

Miyawaki et al. [31] analyzed 24,526 Japanese adults (16% are aged 70 or older) and found that while telemedicine use increased across all age groups from April to September 2020, disparities persisted among older adults. In the 70–79 age group, usage rose from 0.2% to 3.8%. Notably, educational attainment significantly affected uptake: among those aged ≥ 60, individuals with a university degree had higher telemedicine use (6.6%) than those with only a junior high school education (3.5%). These findings suggest that despite increased adoption during the pandemic, low-SES older adults, especially those with limited education, remain at risk of digital exclusion. Addressing digital literacy and accessibility barriers is essential to ensure equitable telehealth integration in aging populations.

Kyaw et al. [32] conducted a cross-sectional study of 323 Korean older adults aged ≥ 65 years to investigate sociodigital disparities in eHealth engagement. Participants with higher education and income levels were significantly more likely to demonstrate high eHealth literacy and use eHealth services. Specifically, education level, monthly income, and digital device ownership were all positively associated with eHealth literacy and usage. These findings highlight that a lower SES among older adults is a major barrier to digital health access, reinforcing the risk of digital exclusion. Addressing disparities in digital literacy and infrastructure is essential to ensure the equitable adoption of telehealth in aging populations.

Although these studies consistently show that a higher SES, such as greater educational attainment, higher income, and digital literacy, is associated with the increased utilization of digital health services among older adults, this association was not observed in a study of the general U.S. population aged 18 to 79 years. Older adults (≥75 years) were significantly less likely to use telemedicine (adjusted OR = 0.63; 95% CI, 0.42–0.94), and those without Internet access also showed reduced use (adjusted OR = 0.62; 95% CI, 0.48–0.81). However, no significant associations were found between telemedicine use and income, education, or race/ethnicity, suggesting that SES did not independently influence overall telemedicine utilization [33]. In this study, older adults accounted for 23.3% of the participants, and the mean age of the study population was 49.4 years. As no subgroup analysis was conducted, the relationship between SES and telemedicine use among older adults remains unclear. However, it is possible that SES exerts a stronger influence on digital health utilization in this age group. Individuals under the age of 40 have grown up with digital technologies as an integral part of daily life, which distinguishes them from older adults. Among the latter, those with a lower SES may still be less inclined to engage with digital health services, suggesting that socioeconomic disparities in utilization persist within the aging population.

Table 2 summarizes the association between SES and the use of digital health among older adults.

### 3.3. Complementary and Alternative Medicine and Supplements

Older adults with a higher SES may be more likely to receive CAM, possibly due to increased health literacy, broader access to information, and the financial capacity to afford non-insurance-covered health services.

According to a nationally representative survey of 7116 U.S. adults aged 65 or older, 29.2% reported using at least one of seven major complementary health approaches [34]. Herbal therapies (18.1%), chiropractic (8.4%), and massage (5.7%) were the most frequently used modalities. Among CAM users, 68.9% perceived CAM as important for maintaining health and well-being, while 52.3% reported overall health improvement and feeling better. Higher education (≥college degree: adjusted OR = 2.40; 95% CI, 1.88–3.07) and higher income (≥400% federal poverty level: adjusted OR = 1.83; 95% CI, 1.50–2.23) were independently associated with greater CAM use. Additionally, older adults with two or more chronic conditions (adjusted OR = 1.39; 95% CI, 1.13–1.73) and functional limitations (adjusted OR = 1.28; 95% CI, 1.09–1.50) were significantly more likely to use CAM.

Higher SES, as indicated by education and income, was significantly associated with greater use of biologically based therapies and mind–body interventions. However, older adults (≥65 years) showed attenuated associations between chronic conditions and CAM use compared to midlife adults. These findings suggest that CAM is more frequently utilized as a self-management strategy by socioeconomically advantaged individuals, yet age-related beliefs about illness may mitigate such usage in later life [35].

Older adults’ attitudes toward CAM appear to differ by ethnicity. Arcury et al. [36] reported that 27.7% of older U.S. adults use CAM, with the highest prevalence among Asians (48.6%), followed by Hispanics (31.6%), Whites (27.7%), and Blacks (20.5%). Compared to Whites, older Asian adults had significantly greater odds of using any CAM (OR = 2.37; 95% CI, 1.41–3.99), particularly alternative systems, biologically based therapies, and mind–body medicine, but lower odds of using body-based methods. Higher education (college graduate vs. less than high school: OR = 3.09; 95% CI, 2.47–3.87) also predicted greater CAM use.

There is a trend indicating that older adults with a higher SES are more likely to use CAM. However, unlike conventional medical consultations, CAM services are mostly self-referred and self-financed and are therefore expected to be strongly influenced by patients’ beliefs and health literacy [37]. It is important to recognize that CAM utilization differs fundamentally from that of modern medical healthcare.

Masumoto et al. [38] conducted a cross-sectional study to investigate factors associated with the use of dietary supplements and over-the-counter medications among older Japanese adults with chronic diseases. The study enrolled 729 outpatients aged ≥ 65 years from a Tokyo hospital. Of these, 32.5% reported the regular use of nonprescription medications, with dietary supplements being most common (28.0%). Vitamins/minerals, aojiru (green juice), and chondroitin–glucosamine were frequently used supplements. Female sex (adjusted OR = 1.58; 95% CI, 1.03–2.41), education beyond high school (OR = 1.70; 95% CI, 1.16–2.49), and better self-rated economic status (OR = 1.62; 95% CI, 1.07–2.43) were significantly associated with nonprescription medication use. Specifically for dietary supplements, female sex (OR = 1.61; 95% CI, 1.03–2.51) and higher education (OR = 1.73; 95% CI, 1.16–2.58) remained significant, while economic status showed a non-significant trend (OR = 1.47; 95% CI, 0.96–2.24). Psychological factors such as anxiety and depression were not associated with supplement use. Notably, 12.2% of participants used nonprescription medications concurrently with ≥5 prescription drugs, highlighting a risk for drug–drug interactions. However, only 30.4% of users disclosed their supplement use to physicians. The authors concluded that supplement use in older Japanese adults is prevalent and influenced by SES, particularly sex, education, and economic perceptions.

Luo et al. [39] conducted a large-scale cross-sectional study examining nutritional supplement (NS) use among older Chinese adults (aged ≥ 60 years), based on data from the 2018 China Health and Retirement Longitudinal Study. Among 11,089 participants, only 0.71% reported using NSs, indicating a significantly low prevalence in this population. SES variables were significantly associated with NS use. Compared with those with no formal education, individuals with a primary or middle school education had 1.32 times higher odds of using any NS (95% CI, 1.14–1.52), while those with high school or higher education had even greater odds (OR = 1.56; 95% CI, 1.25–1.94). A similar gradient was seen for specific supplements such as multivitamins (OR = 2.16; 95% CI, 1.52–3.06) and vitamin A/D (OR = 3.91; 95% CI, 2.55–5.99). Living standard, another SES indicator, was inversely related to NS use. Compared to those with a “good” living standard (reference), those reporting “fair” and “bad/very bad” living conditions had lower odds of NS use: OR = 0.64 (95% CI, 0.56–0.73) and OR = 0.42 (95% CI, 0.32–0.55), respectively. These associations were consistently observed across various types of supplements. Urban residence was also a significant predictor (OR = 1.25; 95% CI, 1.09–1.44), further underscoring geographic SES-related disparities. In summary, this study highlights that while NS use remains rare among older Chinese adults, it is significantly associated with a higher SES, particularly education level, self-perceived economic standard, and urban residency. These findings point to underlying health access inequities in China’s aging population that require targeted policy interventions.

Table 3 summarizes the association between SES and the use of CAM and supplements among older adults.

### 3.4. Lifestyle

There is no doubt that lifestyle factors such as diet, PA, and sleep are paramount for maintaining a healthy life. For instance, SES significantly influences behavioral patterns like excessive alcohol consumption and drug use [40]. Moreover, a higher SES has been associated with more robust social relationships and more effective stress coping mechanisms, as reported in previous studies [41,42]. In this section, the author shall present several pivotal recent studies that have investigated the relationships between dietary habits, PA, sleep, and SES in older adults and synthesize the current state of knowledge.

#### 3.4.1. Diet

Tay et al. [43] investigated sociodemographic and health-related predictors of diet quality, assessed via the Diet Quality Index-International, in 468 community-dwelling pre-frail older adults in New Zealand. The authors identified two independent predictors of lower diet quality: higher body mass index (BMI) (β = −0.17) and residence in areas with high deprivation (low vs. high deprivation: β = 2.14). This study elucidates that SES, as reflected by the New Zealand Deprivation Index, was significantly associated with lower adequacy scores—a subcomponent encompassing the intake of vegetables, fruits, grains, protein, iron, calcium, and vitamin C. These findings emphasize the structural inequities in nutritional access and the critical role of SES in shaping dietary patterns among older adults.

A nationwide cross-sectional study of 3184 older adults in Japan examined the relationship between SES and the frequency of balanced meal consumption [44]. SES was operationalized via two components: educational attainment and subjective financial status (SFS). Compared to older adults with a higher education, those with a lower education showed a significantly higher prevalence of low balanced meal frequency (prevalence ratio (PR) = 1.14; 95% CI, 1.00–1.30). Similarly, older adults reporting poor SFS had a markedly higher prevalence of low balanced meal consumption compared to those with good SFS (PR = 1.24; 95% CI, 1.09–1.43). The interaction analysis further showed that those with both a low education and poor SFS had a significantly elevated risk (PR = 1.76; 95% CI, 1.43–2.17), with a statistically significant overall interaction effect. These findings highlight the compounding burden of educational and economic disadvantage on diet quality in older Japanese adults and emphasize the importance of addressing both the material and educational dimensions of SES to promote nutritional equity in later life.

In the study by Geigl et al. [45] involving older German adults aged 65 to 98 years, a stratified analysis showed significant associations between SES and dietary risk behaviors. The prevalence of low vegetable and fruit intake was higher in the low-SES group (69%) compared to the high-SES group (61%), though this difference did not reach statistical significance. For dairy product consumption, 84% of individuals in the low-SES group had low intake, compared to 76% in the high-SES group, with a marginally significant difference. Whole-grain consumption did not differ significantly across SES groups. Consumption frequency data showed that the mean vegetable/fruit intake frequency was lower in the low-SES group than in the high-SES group. Similarly, dairy consumption frequency was lower in the low-SES group compared to the high-SES group. No significant SES-related differences were observed in whole-grain product consumption. These findings suggest that a lower SES in older adults is associated with a higher prevalence of dietary risk behaviors, particularly in relation to the reduced consumption of fruits, vegetables, and dairy products.

It is understandable that older adults with a higher SES tend to consume higher-quality diets. Foods that are considered health-promoting, such as fresh fruits, vegetables, and lean protein sources, are often more expensive, rendering them inaccessible for routine consumption without a certain level of financial status. Furthermore, appropriate nutritional choices require accurate knowledge regarding the health benefits of various food items, and such knowledge is closely linked to educational attainment. Accordingly, SES is significantly associated with dietary behaviors and nutritional adequacy in later life.

#### 3.4.2. Physical Activity

A cross-sectional analysis from the OUTDOOR ACTIVE study examined PA indicators across SES quintiles among 1507 community-dwelling older adults aged 65–75 years in Germany [46]. SES was derived from income, education, and occupational status. Both self-reported and accelerometer-based PA metrics were assessed. Self-reported total PA (in hours/week) showed a clear SES gradient. Among women, the mean PA declined from 13.27 h/week in the lowest SES quintile to 4.76 h/week in the highest quintile. Among men, a similar decrease was observed, from 13.73 to 5.44 h/week. MVPA also decreased with SES, from 6.61 to 3.14 h/week in women and from 9.23 to 4.39 h/week in men. Correspondingly, the total weekly energy expenditure (METs/week) declined from 49.63 to 20.88 in women and from 56.71 to 25.56 in men. Linear regression analyses adjusted for age and health status demonstrated significant negative associations between SES and PA: per quintile increase in SES, women showed reductions of −0.22 h/week in total PA (95% CI, −0.29 to −0.17), −0.09 h/week in MVPA (95% CI, −0.13 to −0.05), and −0.76 METs/week (95% CI, −1.04 to −0.49). Men exhibited comparable reductions of −0.23, −0.15, and −0.90, respectively. In contrast, accelerometer-derived total PA (counts per minute) did not show a consistent SES gradient among women. In men, however, a positive SES trend was observed: total PA increased from 1497.38 in the lowest quintile to 1624.98 in the highest, with a significant regression coefficient of 2.73 (95% CI, 0.31–5.14). In addition, SES was inversely associated with time spent on specific activities such as housework, gardening, biking, walking, and exercise across both sexes.

Is a higher SES associated with a negative impact on PA in older adults? In fact, the relationship between SES and PA remains inconclusive, with findings that are often contradictory. A systematic review examining PA during the life transition of retirement reported that individuals with a lower SES tend to increase their PA after retirement, whereas those with a higher SES, who are active prior to retirement, show little change in their PA levels following retirement [47].

Kheifets et al. [48] reported strong associations among leisure time physical activity (LTPA), SES, and frailty risk. Among 601 participants, those who were sufficiently active at baseline exhibited a significantly higher SES across multiple dimensions. They attained more years of education (12.4 ± 4.3 years) compared to the inactive groups (8.4 ± 5.6). A larger proportion of the sufficiently active group also reported a high household income (40.9%) relative to the inactive group (34.6%). Furthermore, they resided in more advantaged neighborhoods, as reflected by higher neighborhood SES scores. While education and neighborhood SES showed consistent stepwise increases across PA categories, the difference in income between the sufficiently active and insufficiently active groups remained relatively small. This suggests that once a basic level of financial status is achieved, other SES-related factors, such as educational level and neighborhood context, may play a more decisive role in sustaining higher levels of LTPA among older adults.

#### 3.4.3. Sleep

A narrative review synthesized three decades of empirical research and concluded that low SES is consistently associated with poor sleep health, including short sleep duration, insomnia, poor sleep quality, and excessive daytime sleepiness [49]. These disparities are observed across various SES indicators such as education, income, and employment. To improve sleep-related health inequities, it is essential to address social determinants such as environmental stressors, irregular schedules, and limited access to healthcare. The author introduces a few studies that investigated the relationship between sleep and SES, specifically in older adults.

Xue et al. [50] investigated the association among SES, sleep quality, and multimorbidity in a cohort of 3250 older adults. Compared to those with a very high SES, individuals with a very low SES had significantly higher odds of multimorbidity, with an adjusted OR of 1.440 (95% CI, 1.083–1.913). Poor sleep quality was independently associated with increased odds of multimorbidity (adjusted OR = 2.445; 95% CI, 2.043–2.927). Notably, those with both a very low SES and poor sleep quality demonstrated a higher risk (adjusted OR = 2.787; 95% CI, 1.926–4.034). These results indicate the compounded impact of socioeconomic disadvantage and sleep impairment on multimorbidity risk, suggesting the need for integrated interventions targeting both social determinants and sleep health in older populations.

Collinge et al. [51] analyzed data from 7040 older adults in the ELSA COVID-19 sub-study and found that self-reported poor sleep quality was significantly associated with financial hardship and concerns about future finances. Specifically, individuals who were “very worried” or “extremely worried” about future finances had markedly elevated odds of poor sleep (OR = 2.02; 95% CI, 1.15–3.55; OR = 8.09; 95% CI, 1.62–40.30, respectively). These associations remained significant after adjusting for sociodemographic, mental health, and physical health factors, indicating that financial anxiety is an independent risk factor for impaired sleep in this population. It is noteworthy that sleep quality remained significantly associated with financial concerns even after adjusting for physical and mental health conditions; however, SES was not objectively assessed, and the data were collected during the COVID-19 pandemic, which may have influenced the study results.

Table 4 summarizes the association between SES and lifestyle among older adults.

## 4. Discussion

SES is a strong determinant of healthcare utilization patterns among older adults, influencing not only access to conventional medical services but also engagement with digital health, CAM, diet, PA, and sleep health. Across diverse global contexts, a higher SES—reflected by higher income, education, and health literacy—is consistently associated with increased access to healthcare and preventive services, as well as healthier behaviors such as balanced diets, daily PA, and better sleep quality. In contrast, a lower SES correlates with heightened risks of both underuse and inappropriate overuse of services, digital exclusion, and lifestyle-related vulnerabilities, regardless of clinical need. Importantly, region- and system-level disparities further compound individual-level inequities, underscoring the critical role of structural determinants in shaping older adults’ health trajectories. Bridging these gaps is essential for achieving equity in healthy aging.

Addressing health disparities among older adults caused by SES and ensuring fair and appropriate medical care for those in need remains a significant challenge. It is also difficult to change individual factors such as educational level or personal preferences; thus, addressing disparities will, in principle, require the enhancement of social support mechanisms. There is a randomized controlled trial that investigated whether providing financial assistance to low-income older adults would affect patient outcomes and medication adherence. In this study, copayments (approximately 30%, averaging 35 Canadian dollars per month) for 15 classes of cardiovascular medications were entirely eliminated for adults aged 65 and older with high cardiovascular risk and an annual income below 50,000 Canadian dollars. While statin adherence improved from 69% to 72%, there were no significant reductions in the risks of cardiovascular events, mortality, or hospitalization [52]. An analysis of eight studies that provided transportation subsidies to improve access to medical care demonstrated reductions in appointment no-show rates and increased follow-up visit attendance. However, few rigorously designed studies have evaluated the impact of transportation services on health outcomes or healthcare utilization. Existing evidence suggests that such interventions may be more effective when integrated with broader efforts to address social and economic barriers to care [53]. A systematic review showed that community-based health worker interventions, particularly those involving tailored education, care coordination, and support for self-management, were associated with modest improvements in clinical outcomes such as glycemic control (e.g., mean HbA1c reduction of approximately 0.3 to 0.5 percentage points in several studies) and blood pressure control [54]. These studies suggest that effecting meaningful changes in health behaviors and improving clinical outcomes among older adults with a low SES require large-scale, comprehensive social support, including financial assistance, support for accessing healthcare services, and opportunities for health education. However, in aging societies, rising social security expenditures may place considerable strain on public finances, potentially limiting the feasibility of providing such extensive support. On the other hand, interventions using digital health may be promising. In a study conducted among older adults living in low-income areas on the outskirts of São Paulo, Brazil, home-based psychoeducation delivered via a tablet application led to a significant improvement in depression (adjusted OR = 2.16: 95% CI, 1.47–3.18) [55]. In another study targeting rural and low-income populations in the United States, the use of a telehealth device that uploaded daily blood glucose and blood pressure readings to a central server, combined with biweekly remote medication adjustments by nurses, resulted in a marked improvement in glycemic control (6-month change in HbA1c: −0.99% in the intervention group vs. control) [56]. While this narrative review has noted the limited digital literacy among older adults with a low SES, it is possible that providing digital health education along with necessary devices free of financial burden could improve health behaviors in this population and generate a positive public health impact. According to a recent scoping review, eHealth interventions held considerable promise for improving health behaviors among individuals with a low SES; however, their effectiveness was contingent upon addressing both technological and social barriers [57]. Although enhancing digital literacy among older adults with a low SES remained challenging, previous studies demonstrated that the use of simplified user interfaces, the provision of robust peer support through family and community networks, the active involvement of healthcare professionals, and the adoption of co-design approaches that incorporated the perspectives of the target population facilitated positive behavioral change [57]. In addition, other evidence indicated that the implementation of health promotion programs contributed to improvements in nutritional status and instrumental ADL among older adults from socioeconomically disadvantaged backgrounds [58]; enrollment in public pension schemes was associated with enhanced physical functioning and self-rated mental health in older adults living in rural areas [59]; and participation in weekly health promotion activities was shown to promote active aging, defined as the process of enhancing health, ensuring social security, and fostering social engagement among older individuals [59]. Collectively, these findings indicated the critical role of governmental and healthcare sector leadership in developing and implementing targeted interventions to improve the health and well-being of older adults with a low SES.

However, interventions targeting older adults do not always produce beneficial outcomes. For example, a digital literacy improvement program for older adults with a low SES successfully enhanced participants’ digital skills but did not lead to improvements in loneliness, social connectedness, QOL, or subjective well-being [60]. Even if digital literacy improves, it may not necessarily result in sustained use or translate into tangible health benefits. Similarly, a home-based health promotion program for older adults was reported to have no significant impact on functional ability or QOL [61]. Although these interventions involved in-person visits to older adults’ homes, both human and financial resources are limited. In an increasingly aging society, it is possible that such resource-intensive health promotion programs may not be feasible at scale due to resource constraints.

The World Health Organization and the United Nations have launched the Decade of Healthy Ageing (2021–2030), which aims to ensure that all older adults can live longer and healthier lives. This initiative promotes four key action areas: changing societal attitudes toward aging, creating age-friendly communities that enable older adults to flourish, delivering integrated and person-centered health services, and ensuring equitable access to long-term care [62]. However, the lack of adequate empirical data on the realities faced by older adults raises concerns about the feasibility of these goals. Therefore, in order to achieve sustainable and effective health promotion for older adults, it is essential to advance research on the relationship between SES and health behaviors in this population, accumulate relevant data, and develop strategies for efficient and targeted resource allocation.

It is also important to note that SES is influenced by external factors such as the historical period and economic conditions. This review introduced studies conducted during the COVID-19 pandemic that examined the relationship between SES and sleep among older adults. Are there any differences compared to findings from before the pandemic? According to a review summarizing observational studies conducted between 1990 and 2019, a lower SES, such as lower income or education, was associated with poorer sleep continuity and quality among older adults, consistent with findings from studies conducted during the pandemic [49]. On the other hand, a study reported that during the pandemic, approximately half of older adults experienced deteriorated sleep quality, 40% had short sleep duration, and 20% reported insomnia symptoms [63], suggesting that the relationship between SES and sleep may have been exaggerated during the pandemic. Nevertheless, the association between SES and sleep has shown consistency, and there is no doubt that addressing sleep problems among older adults with a low SES can contribute to healthy aging.

SES, commonly measured by education, income, and occupation, reflects an individual’s overall economic and social position and is generally associated with better health outcomes [64]. Older adults scored higher than both young and middle-aged adults in an overall health-promoting lifestyle, particularly in the areas of health responsibility, nutrition, and stress management. However, SES accounted for only 13.4% of the variance in overall lifestyle scores, and between 5.2% and 18.6% of the variance across the six individual lifestyle dimensions [65]. In other words, older adults tend to engage in healthier behaviors than younger individuals, regardless of their SES. It is presumed that as people age and gradually experience a decline in physical strength and health, which is often accompanied by multiple health conditions, they tend to place greater value on their health than they did when they were younger and become more cautious about behaviors that may harm it. Among the studies included in this narrative review, a few studies reported that comorbidities influenced the relationship between SES and health behaviors in older adults [34,50]. These characteristics of older individuals should be taken into account when interpreting the research findings. Therefore, it should be noted that among older adults, while disease burden may enhance motivation to engage in health-promoting behaviors, limited resources, particularly financial ones, associated with a low SES may hinder such behaviors, resulting in a motivation–resource gap. Moreover, it should be considered that individuals who have consistently engaged in healthy behaviors are more likely to survive into older age and remain within higher SES groups. Thus, even within the category of older adults, the circumstances may differ depending on whether one is in their 60s, 70s, or 80s. Although this is beyond the scope of the present review, factors such as gender, ethnicity, and regionality, apart from age, may interact with SES [66,67,68]; thus, it is necessary to incorporate these factors into the analysis as well. Furthermore, in older adults, subjective economic status and a sense of social roles may play a more significant role in determining health behaviors than income or employment status [69]. Indeed, several studies have reported that higher subjective SES is associated with reduced risks of mortality, as well as declines in ADL and cognitive function [70,71,72]. Subjective SES has been shown to exhibit a unique cumulative association with physical health in adults, over and above traditional objective indicators of SES [73,74]. Furthermore, there are reports that subjective SES, particularly subjective evaluations of educational attainment and occupational status, is more strongly associated with well-being than conventional SES measures and serves as a significant predictor of psychological well-being [75]. The relationship between subjective SES and health behaviors in older adults remains insufficiently studied. Although subjective SES is beyond the scope of this review, future research should investigate the relationship between the level of subjective SES and health behaviors among older adults.

This narrative review highlighted a substantial body of evidence from Japan. One reason is that Japan is the most rapidly aging country in the world. However, in other countries such as the United States, the United Kingdom, Canada, Australia, Scotland, and Sweden, it has also been reported that individuals with a lower SES are more likely to engage in unhealthy behaviors such as smoking, physical inactivity, and poor diet [76]. Moreover, according to a study by Tanaka et al. [77], individuals with a lower SES were more likely to report poor health in most countries, indicating that socioeconomic inequality significantly affects health. In particular, health disparities based on educational attainment were pronounced in European countries and also marked in the United States. In contrast, in Japan, health disparities were more strongly associated with income levels. The authors suggested that these differences may be influenced by variations in welfare systems, educational structures, cultural contexts, and access to healthcare across countries. Therefore, while the findings of this review may have a certain degree of generalizability, it is important to consider cross-national differences in cultural backgrounds and healthcare systems.

This narrative review has several limitations. First, it was conducted by a single author, which may have introduced bias in the selection and evaluation of studies, potentially resulting in the omission of important research. To mitigate this limitation, future systematic reviews should involve multiple authors who independently perform literature searches and assessments. Second, the purpose of this narrative review was not to provide a comprehensive overview of all studies examining the relationship between SES and health behaviors in older adults, and databases other than PubMed were not searched. To capture a broader range of relevant studies, future reviews should incorporate additional databases such as Embase, Scopus, or AMED. Third, this review focused on the specific domains of health behaviors; namely, healthcare utilization, digital health, the use of CAM, and lifestyle factors. However, health behaviors encompass a wide range of other activities, such as infection prevention practices, substance abuse, alcohol and tobacco use, and mental health management. Although it is difficult to comprehensively cover all aspects, examining each of these behaviors individually would contribute to a more nuanced understanding of the relationship between SES and health behaviors in older adults. Fourth, this review largely focuses on associations rather than underlying mechanisms or causality; future iterations should more explicitly address potential mediating and moderating factors (e.g., health literacy, access barriers, social capital). Ideally, evidence from intervention studies such as randomized controlled trials would be preferred; however, interventions targeting SES are virtually impossible, and even if some forms of intervention were implemented, blinding would be difficult. Therefore, a carefully designed study framework is required. In practice, it would be more realistic to analyze large-scale, long-term observational studies to identify mediating and moderating factors.

## 5. Conclusions

The evidence synthesized across clinical studies converges on a lesson: SES is a key determinant of older adults’ engagement across all major domains of health behavior. Therefore, a sustainable healthcare system for super-aged societies must regard SES not as a peripheral covariate but as a core design factor. By coupling redistributive financing with digital technology, aging societies can transform the current SES-stratified landscape into one in which healthy longevity is both attainable and economically sustainable. Rapid improvement in the SES of older adults is unrealistic, and disparities among individuals are inevitable. To address this, governments and local authorities should strengthen fiscal policies and expand social support systems aimed at raising the overall SES baseline. In parallel, organizations involved in health-related services must maintain ongoing efforts to educate and empower patients, thereby fostering healthier behaviors, particularly among those with a lower SES (Figure 1).

## Figures and Tables

**Figure 1 healthcare-13-01669-f001:**
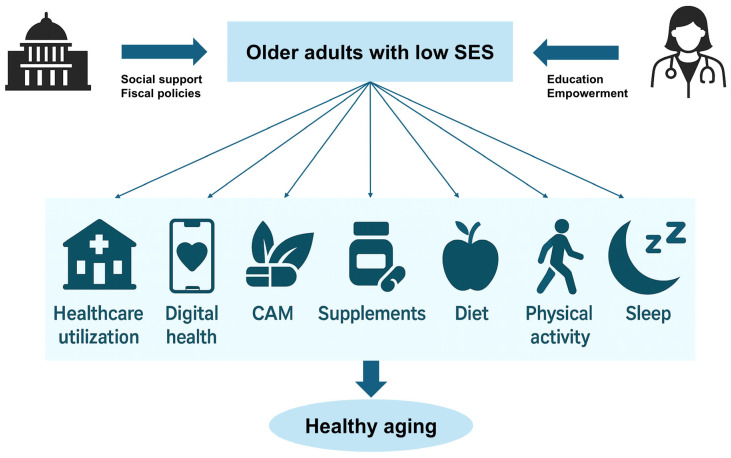
By continuously engaging older adults with low socioeconomic status, it is possible to achieve a society that promotes healthy aging.

**Table 1 healthcare-13-01669-t001:** Association between socioeconomic status and healthcare utilization among older adults.

Author, Year	Study Design	Participants	Variables	Results
Hoeck et al., 2011 [18]	Cross-sectional study	Non-institutionalized adults aged ≥ 65 years (*n* = 4494); Belgium	SES (income, education, housing tenure) and general practitioner/specialist contacts	No significant SES differences in general practitioner/specialist visits after adjustment for health and enabling factors
Mai et al., 2022 [19]	Cross-sectional study (Chinese Longitudinal Healthy Longevity Survey 2018)	Older adults aged ≥ 65 years with limited ADL (*n* = 3980); China	Economic status and access to healthcare services	OR = 2.98 (fair vs. poor), OR = 7.23 (rich vs. poor) for access to care; OR = 2.33 for daily life affordability
Sun et al., 2024 [21]	Cross-sectional study using nationwide Japanese data	Older adults aged ≥ 65 years across 333 secondary medical areas; Japan	Medical/long-term care resources, population density, SES proxies vs. home-visit rates	Coefficient = 0.21 (enhanced home care support clinics/ hospitals), coefficient = 0.17 (conventional home care support clinics/ hospitals), coefficient = 0.10 (population density)
Wachelder et al., 2017 [22]	Retrospective cohort study	Community-dwelling older emergency department patients aged ≥ 65 years (*n* = 4828); Netherlands	SES (income at postal code level) and outcomes: hospitalization, mortality, 30-day revisit	Low SES associated with higher hospitalization risk (adjusted OR = 1.3); no SES effect on other outcomes
Leff et al., 2023 [23]	Cohort study using Medicare Advantage claims (2017–2018)	Medicare Advantage beneficiaries aged ≥ 65 years (*n* = 38,800 home-based medical care users vs. 132,147 controls); USA	SES and use of home-based medical care	Lower SES associated with higher home-based medical care use (data from income-linked claims)
Tu and Cohen, 2008 [25]	Cross-sectional study (Health Tracking Survey 2007)	Adults aged ≥ 18 years, including older adults aged ≥ 65	Education level and health information seeking behavior	Information seeking: 56% overall in 2007; Internet use rose from 16% (2001) to 32% (2007); SES gradient evident
Choi, 2011 [26]	Cross-sectional study (US National Health Interview Survey 2009)	Older adults aged ≥ 65 years (*n* = 5294); USA	SES and use of health IT	Health IT use: 32.2% (65–74 years), 14.5% (75–84 years), 4.9% (≥85 years); higher SES significantly associated with greater health IT use

SES, socioeconomic status; ADL, activities of daily living; OR, odds ratio; IT, information technology.

**Table 2 healthcare-13-01669-t002:** Association between socioeconomic status and the use of digital health among older adults.

Author, Year	Study Design	Participants	Variables	Results
Fealy et al., 2023 [29]	Repeated cross-sectional study	Older adults aged ≥ 65 years (*n* = 21,830); Australia	Socioeconomic disadvantage (area-level) vs. telehealth use	Higher telehealth use in most disadvantaged areas (deciles 1–3); SES positively associated with use in early pandemic phase
Gordon and Hornbrook, 2016 [30]	Survey and database study	Older adults aged ≥ 65 years; USA	Race/ethnicity, income, education vs. access to and preferences for patient portals and eHealth technologies	Black and Hispanic older adults had lower portal access; lower income and education associated with less preference for digital tools
Miyawaki et al., 2021 [31]	Cross-sectional study	Adults aged 18–79 years (*n* = 24,526); Japan	Education, income, urbanicity vs. telemedicine use	Higher education and urban living associated with more telemedicine use; income not significantly associated
Kyaw et al., 2024 [32]	Cross-sectional study	Community-dwelling older adults aged ≥ 65 years (*n* = 434); Korea	Income and social media use vs. eHealth literacy and eHealth use	Higher income and social media use (OR = 3.97) associated with greater eHealth use
Chang et al., 2024 [33]	Cross-sectional study	Adults aged ≥18 years, including older adults aged ≥ 65 (*n* = 5437); USA	Income and education vs. telemedicine use and type (video vs. audio)	No significant differences in telemedicine use or modality by income or education

SES, socioeconomic status; OR, odds ratio.

**Table 3 healthcare-13-01669-t003:** Association between socioeconomic status and the use of complementary and alternative medicine and supplements among older adults.

Author, Year	Study Design	Participants	Variables	Results
Rhee et al., 2018 [34]	Cross-sectional study (2012 National Health Interview Survey)	Older adults aged ≥ 65 (*n* = 7116); USA	SES and use of 7 major CAM therapies	29.2% used CAM; higher education (OR = 2.40), income (OR = 1.83), chronic illness (OR = 1.39), functional limitations (OR = 1.28) linked to CAM use
Grzywacz et al., 2007 [35]	Cross-sectional	Older adults aged ≥ 65; USA	Age, ethnicity, chronic disease, and CAM use	Higher SES linked to biologically based and mind–body therapies; illness-related CAM use lower in older vs. midlife adults
Arcury et al., 2006 [36]	Cross-sectional study (2002 National Health Interview Survey)	Older adults aged ≥ 65 (*n* = 701); USA	Ethnicity, education, and CAM use	CAM use highest in Asians (48.6%); higher education (OR = 3.09) increased CAM use; ethnic differences in modality preference
Masumoto et al., 2018 [38]	Cross-sectional	Older adults aged ≥ 65 with chronic diseases (*n* = 729); Japan	SES and use of dietary supplements and over-the-counter medications	32.5% used over-the-counter medications; associated with female sex (OR = 1.61), higher education (OR = 1.73), better economic status (OR = 1.47); 30.4% disclosed use
Dong et al., 2022 [39]	Cross-sectional study (2018 Chinese Longitudinal Healthy Longevity Survey)	Older adults aged ≥ 65 (*n* = 11,089); China	SES and use of nutritional supplements	0.71% used supplements; linked to female sex (OR = 1.71), urban hukou (OR = 1.25), higher education (OR = 1.56), better living standard

SES, socioeconomic status; CAM, complementary and alternative medicine; OR, odds ratio.

**Table 4 healthcare-13-01669-t004:** Association between socioeconomic status and lifestyle among older adults.

Author, Year	Study Design	Participants	Variable	Results
Tay et al., 2023 [43]	Cross-sectional study	Older adults aged ≥ 65 years with pre-frailty (*n* = 468); New Zealand	SES and diet quality	Higher income (β = 0.12) and home ownership (β = 0.13) associated with better diet quality
Nishinakagawa et al., 2023 [44]	Cross-sectional study	Adults aged 20–79 years including older adults aged ≥ 65 (*n* = 8464); Japan	Education and financial status vs. dietary habits	Among ≥ 65, low education (OR = 0.48) and poor financial status (OR = 0.42) associated with low vegetable intake
Geigl et al., 2022 [45]	Cross-sectional study	Older adults aged ≥ 65 years (*n* = 1687); Germany	Education, income, former occupation vs. dietary risk behavior	Low SES (esp. education) associated with dietary risk behavior
Stalling et al., 2022 [46]	Cross-sectional study	Older adults aged 65–75 years (*n* = 1507); Germany	Education and income vs. physical activity	Lower SES significantly associated with reduced odds of sufficient physical activity (OR = 0.54 for lowest vs. highest SES)
Kheifets et al., 2022 [48]	Prospective cohort study	Older adults aged ≥ 65 years (*n* = 1799 at baseline, *n* = 601 at follow-up); Israel	Education, income, neighborhood SES vs. physical activity and frailty	Low SES linked to less physical activity; inactivity associated with frailty (OR = 2.77)
Xue et al., 2022 [50]	Cross-sectional study	Older adults aged > 60 years (*n* = 3250); China	SES and sleep quality vs. multimorbidity	Very low SES (OR = 1.44) and poor sleep (OR = 2.45) linked to higher multimorbidity; combined OR = 3.14
Collinge and Bath, 2023 [51]	Cross-sectional study	Older adults aged ≥ 50 years (*n* = 7040); England	Education and financial situation vs. sleep quality	Lower education and financial hardship associated with poorer sleep quality

SES, socioeconomic status; OR, odds ratio.

## Data Availability

The data that support the findings of this study are available from the author upon reasonable request.

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
