# Peer review of "The Relationship Between Socioeconomic Status and Health Behaviors in Older Adults: A Narrative Review"

_healthcare, 2025, doi:10.3390/healthcare13141669_

Round 1
Reviewer 1 Report
Comments and Suggestions for Authors
Thank you for the opportunity to review this manuscript entitled "The Relationship Between Socioeconomic Status and Health Behaviors in Older Adults." This narrative review addresses a topic of high relevance in the context of rapidly aging societies, with a focus on Japan as a leading example. The manuscript synthesizes evidence on how socioeconomic status (SES) shapes health behaviors among older adults, discusses the implications for health equity, and suggests potential policy directions. Below, I provide a detailed evaluation organized by key criteria.
1. Significance and Originality
While the topic is significant, the narrative approach limits the opportunity for meta-analytic synthesis or systematic quantification of SES gradients.
The originality could be enhanced by a more explicit comparison with evidence from non-Japanese or non-Asian contexts, or by addressing how Japanese findings may inform broader international policy.
2. Methodological Rigor
The narrative review design is inherently limited in terms of reproducibility and potential selection bias. Consideration of a systematic review or at least a PRISMA-style flow diagram would strengthen transparency.
Search parameters (databases, time frame, inclusion/exclusion criteria) should be described in more detail for replicability.
There is insufficient discussion of study quality assessment or risk of bias in the included articles.
3. Results and Interpretation
The results would benefit from tabulated summaries of key studies, SES measures, and outcomes to enhance clarity.
The review largely focuses on the association rather than underlying mechanisms or causality, future iterations should more explicitly address possible mediating and moderating factors (e.g., health literacy, access barriers, social capital).
Some sections are heavily weighted towards Japanese data; consideration of generalizability beyond Japan would be valuable.
4. Discussion and Policy Implications
The discussion could be further deepened by addressing potential interventions with evidence of effectiveness, such as tailored health promotion or digital literacy programs for low-SES older adults.
Consideration of potential unintended consequences or challenges in implementing SES-targeted interventions in aging societies would enhance the critical depth.
Reference to international policy frameworks (e.g., WHO Decade of Healthy Ageing) would situate the discussion in a global context.
5. Presentation, Language, and Formatting
There are several minor language and typographical errors throughout the manuscript; careful proofreading and language editing are recommended to improve clarity and readability.
The abstract is dense and would benefit from clearer sectioning and more concise statements of the main findings and recommendations.
Figures or tables summarizing evidence would enhance the manuscript’s accessibility for readers.
Specific Recommendations:
Provide more detailed description of the search strategy, inclusion/exclusion criteria, and study quality assessment.
Include tables or figures to summarize key findings and studies.
Address the limitations of the narrative approach and consider how findings generalize beyond Japan.
Expand the discussion of policy implications and link findings to international frameworks.
Conduct thorough language editing to address typographical and grammatical errors.
Thank you again for the opportunity to review this important and timely manuscript.
Author Response
Dear Reviewer 1,
Comment: Thank you for the opportunity to review this manuscript entitled "The Relationship Between Socioeconomic Status and Health Behaviors in Older Adults." This narrative review addresses a topic of high relevance in the context of rapidly aging societies, with a focus on Japan as a leading example. The manuscript synthesizes evidence on how socioeconomic status (SES) shapes health behaviors among older adults, discusses the implications for health equity, and suggests potential policy directions. Below, I provide a detailed evaluation organized by key criteria.
Reply: Thank you for taking the time to review the manuscript. I have revised the manuscript in accordance with your comments and suggestions as follows.
- Significance and Originality
Comment: While the topic is significant, the narrative approach limits the opportunity for meta-analytic synthesis or systematic quantification of SES gradients.
Reply: I agree with your comment; this is a major limitation of the review. I have addressed this point in the study limitation section. “First, it was conducted by a single author, which may have introduced bias in the se-lection and evaluation of studies, potentially resulting in the omission of important research. To mitigate this limitation, future systematic reviews should involve multiple authors who independently perform literature searches and assessments. Second, the purpose of this narrative review was not to provide a comprehensive overview of all studies examining the relationship between SES and health behaviors in older adults, and databases other than PubMed were not searched. To capture a broader range of relevant studies, future reviews should incorporate additional databases such as Em-base, Scopus, or AMED.”
Comment: The originality could be enhanced by a more explicit comparison with evidence from non-Japanese or non-Asian contexts, or by addressing how Japanese findings may inform broader international policy.
Reply: I have addressed this issue and revised the Discussion section accordingly. “This narrative review highlighted a substantial body of evidence from Japan. One reason is that Japan is the most rapidly aging country in the world. However, in other countries such as the United States, United Kingdom, Canada, Australia, Scotland, and Sweden, it has also been reported that individuals with lower SES are more likely to engage in unhealthy behaviors such as smoking, physical inactivity, and poor diet [76]. Moreover, according to a study by Tanaka et al. [77], individuals with lower SES were more likely to report poor health in most countries, indicating that socioeconomic inequality significantly affects health. In particular, health disparities based on educational attainment were pronounced in European countries and also marked in the United States. In contrast, in Japan, health disparities were more strongly associated with income levels. The authors suggested that these differences may be influenced by variations in welfare systems, educational structures, cultural contexts, and access to healthcare across countries. Therefore, while the findings of this review may have a certain degree of generalizability, it is important to consider cross-national differences in cultural backgrounds and healthcare systems.”
- Methodological Rigor
Comment: The narrative review design is inherently limited in terms of reproducibility and potential selection bias. Consideration of a systematic review or at least a PRISMA-style flow diagram would strengthen transparency.
Search parameters (databases, time frame, inclusion/exclusion criteria) should be described in more detail for replicability.
There is insufficient discussion of study quality assessment or risk of bias in the included articles.
Reply: Thank you for your thoughtful comments. As you pointed out, a well-designed systematic review can synthesize evidence more scientifically and objectively than a narrative review. However, previous scholarly papers discussing review methodologies have clearly stated that narrative reviews serve educational and informative purposes in scientific journals, and thus are not required to follow PRISMA guidelines or assess the risk of bias in included studies [1,2]. Additionally, a narrative review aims to summarize and synthesize existing literature on a particular topic but does not seek to produce generalizable or cumulative knowledge. Rather, its purpose is to provide readers with a comprehensive background for understanding the current state of knowledge and to highlight the significance of new research [3]. In light of this, I chose to conduct a narrative review rather than a systematic review.
To avoid any misunderstanding by readers, I have added “A Narrative Review” to the title.
According to your suggestion, I have revised the Methods section as follows:
“2.1. Research Question
The following question was formulated to guide this review:
Generally, individuals with higher SES have greater health awareness and literacy and are more likely to engage in healthier behaviors; however, is this also true for older adults?
2.2. Literature Sources
The author performed an electronic literature search on PubMed for articles published up to May 2025.
2.3. Search Parameters
The search terms used were “socioeconomic status”; “health behavior,” “digital health,” “telemedicine,” “complementary therapies,” “diet,” “physical activity,” and “sleep”—all of which are included as Medical Subject Headings (MeSH); and “older adults.” All types of articles were considered from the database’s inception (n = 6,880). Of these, clinical studies, including cross-sectional and observational studies, were deemed eligible (n = 1,098). The titles and abstracts of the identified articles were re-viewed to assess their relevance. The author also reviewed the reference lists of the selected articles to identify further eligible studies.
2.4. Data Cleaning
The article selection criteria included the following: (a) written in English; (b) involving older adults aged 65 years or older; (c) including objectively evaluated SES variables (e.g., education, income, and occupation); and (d) addressing health behaviors such as healthcare utilization, digital health, CAM, supplements, diet, physical activity, and sleep. The exclusion criteria were as follows: (a) article types such as re-views, case reports, editorials, commentaries, letters, and protocols; (b) studies that did not include SES variables; (c) studies without quantitative analysis; and (d) preprints.
A total of 24 studies were included in this review.”
- Results and Interpretation
Comment: The results would benefit from tabulated summaries of key studies, SES measures, and outcomes to enhance clarity.
Reply: I would appreciate it if you would check the summary tables (Table 1-4).
Comment: The review largely focuses on the association rather than underlying mechanisms or causality, future iterations should more explicitly address possible mediating and moderating factors (e.g., health literacy, access barriers, social capital).
Reply: I agree with your suggestion and have added the following text to the Discussion section:
“Forth, this review largely focuses on associations rather than underlying mechanisms or causality; future iterations should more explicitly address potential mediating and moderating factors (e.g., health literacy, access barriers, social capital). Ideally, evidence from intervention studies such as randomized controlled trials would be preferred; however, interventions targeting SES are virtually impossible, and even if some forms of intervention were implemented, blinding would be difficult. Therefore, a carefully designed study framework is required. In practice, it would be more realistic to analyze large-scale, long-term observational studies to identify mediating and moderating factors.”
Comment: Some sections are heavily weighted towards Japanese data; consideration of generalizability beyond Japan would be valuable.
Reply: According to your comment, I have added the following text to the Discussion section:
“This narrative review highlighted a substantial body of evidence from Japan. One reason is that Japan is the most rapidly aging country in the world. However, in other countries such as the United States, United Kingdom, Canada, Australia, Scotland, and Sweden, it has also been reported that individuals with lower SES are more likely to engage in unhealthy behaviors such as smoking, physical inactivity, and poor diet [76]. Moreover, according to a study by Tanaka et al. [77], individuals with lower SES were more likely to report poor health in most countries, indicating that socioeconomic inequality significantly affects health. In particular, health disparities based on educational attainment were pronounced in European countries and also marked in the United States. In contrast, in Japan, health disparities were more strongly associated with income levels. The authors suggested that these differences may be influenced by variations in welfare systems, educational structures, cultural contexts, and access to healthcare across countries. Therefore, while the findings of this review may have a certain degree of generalizability, it is important to consider cross-national differences in cultural backgrounds and healthcare systems.”
- Discussion and Policy Implications
Comment: The discussion could be further deepened by addressing potential interventions with evidence of effectiveness, such as tailored health promotion or digital literacy programs for low-SES older adults.
Reply: Based on your comment, I have added the following text to the Discussion:
“According to a recent scoping review, eHealth interventions held considerable promise for improving health behaviors among individuals with low SES; however, their effectiveness was contingent upon addressing both technological and social barriers [57]. Although enhancing digital literacy among older adults with low SES remained challenging, previous studies demonstrated that the use of simplified user interfaces, the provision of robust peer support through family and community networks, the active involvement of healthcare professionals, and the adoption of co-design approaches that incorporated the perspectives of the target population facilitated positive behavioral change [57]. In addition, other evidence indicated that the implementation of health promotion programs contributed to improvements in nutritional status and instrumental ADL among older adults from socioeconomically disadvantaged back-grounds [58]; enrollment in public pension schemes was associated with enhanced physical functioning and self-rated mental health in older adults living in rural areas [59]; and participation in weekly health promotion activities was shown to promote active aging, defined as the process of enhancing health, ensuring social security, and fostering social engagement among older individuals [59]. Collectively, these findings indicated the critical role of governmental and healthcare sector leadership in devel-oping and implementing targeted interventions to improve the health and well-being of older adults with low SES.”
Comment: Consideration of potential unintended consequences or challenges in implementing SES-targeted interventions in aging societies would enhance the critical depth.
Reply: Based on your suggestion, I have added the following text to the Discussion:
“However, interventions targeting older adults do not always produce beneficial outcomes. For example, a digital literacy improvement program for older adults with low SES successfully enhanced participants' digital skills, but did not lead to improvements in loneliness, social connectedness, QOL, or subjective well-being [60]. Even if digital literacy improves, it may not necessarily result in sustained use or translate into tangible health benefits. Similarly, a home-based health promotion pro-gram for older adults was reported to have no significant impact on functional ability or QOL [61]. Although these interventions involved in-person visits to older adults’ homes, both human and financial resources are limited. In an increasingly aging society, it is possible that such resource-intensive health promotion programs may not be feasible at scale due to resource constraints.”
Comment: Reference to international policy frameworks (e.g., WHO Decade of Healthy Ageing) would situate the discussion in a global context.
Reply: I have referred to an international policy framework.
“The World Health Organization and the United Nations have launched the Decade of Healthy Ageing (2021–2030), which aims to ensure that all older adults can live longer and healthier lives. This initiative promotes four key action areas: changing societal attitudes toward aging, creating age-friendly communities that enable older adults to flourish, delivering integrated and person-centered health services, and ensuring equitable access to long-term care [62]. However, the lack of adequate empirical data on the realities faced by older adults raises concerns about the feasibility of these goals. Therefore, in order to achieve sustainable and effective health promotion for older adults, it is essential to advance research on the relationship between SES and health behaviors in this population, accumulate relevant data, and develop strategies for efficient and targeted resource allocation.”
- Presentation, Language, and Formatting
Comment: There are several minor language and typographical errors throughout the manuscript; careful proofreading and language editing are recommended to improve clarity and readability.
Reply: I have carefully proofread and revised the manuscript.
Comment: The abstract is dense and would benefit from clearer sectioning and more concise statements of the main findings and recommendations.
Reply: I have revised the abstract accordingly.
Comment: Figures or tables summarizing evidence would enhance the manuscript’s accessibility for readers.
Reply: I have added a summary figure to the Conclusion section.
Specific Recommendations:
Provide more detailed description of the search strategy, inclusion/exclusion criteria, and study quality assessment.
Include tables or figures to summarize key findings and studies.
Address the limitations of the narrative approach and consider how findings generalize beyond Japan.
Expand the discussion of policy implications and link findings to international frameworks.
Conduct thorough language editing to address typographical and grammatical errors.
Thank you again for the opportunity to review this important and timely manuscript.
Reply: As noted above, I have revised the manuscript in accordance with your comments and suggestions. I would be grateful if you could kindly review the revised version.
References
- Parums DV. Editorial: Review Articles, Systematic Reviews, Meta-Analysis, and the Updated Preferred Reporting Items for Systematic Reviews and Meta-Analyses (PRISMA) 2020 Guidelines. Med Sci Monit. 2021;27:e934475.
- Baethge C, Goldbeck-Wood S, Mertens S. SANRA-a scale for the quality assessment of narrative review articles. Res Integr Peer Rev. 2019;4:5.
- Davies P The Relevance of Systematic Reviews to Educational Policy and Practice. Oxford Rev Edu. 2000;26(3–4):365–378.
Reviewer 2 Report
Comments and Suggestions for Authors
Dear Authors,
Thank you for providing me with the opportunity to review this interesting paper. Below, I have listed my comments:
1) For the introduction, the paper mostly describes findings but does not critically evaluate them. Are there inconsistencies across studies? Are there limitations in the methods used (e.g., self-reporting, regional biases)?
2) The methods section lacks some transparency. The literature source is only PubMed; why were more excluded sources excluded?
3) There is no information as to how many articles were included/excluded.
4) In the sleep section, the limitations of COVID-era data are acknowledged (line 537), but should be emphasized more in relation to generalizability. Perhaps contrast with pre-pandemic findings if available.
5) In the discussion, the mention of subjective SES is compelling, but underdeveloped. You could expand on how subjective SES might interact with objective SES or other psychosocial factors in influencing health behaviors.
6) While the need to “enhance social support” is mentioned, concrete recommendations are sparse.
7) It might not be measured in the studies or stated but it might be key to mention in the discussion that factors like gender, ethnicity, or rurality often interact with SES. You could briefly acknowledge that SES rarely operates in isolation and intersects with other axes of disadvantage.
I hope this feedback is helpfu.
Author Response
Dear Reviewer 2,
Comment: Thank you for providing me with the opportunity to review this interesting paper. Below, I have listed my comments:
1) For the introduction, the paper mostly describes findings but does not critically evaluate them. Are there inconsistencies across studies? Are there limitations in the methods used (e.g., self-reporting, regional biases)?
Reply: Thank you for your important comment. The studies included in this review objectively assessed SES, such as income and educational attainment, using either continuous or categorical variables. As shown in the results, the reported findings are generally consistent. However, as you pointed out, SES is influenced by factors such as ethnicity and regional characteristics, and the usefulness of subjective SES has also been the subject of discussion. I have revised the Discussion section in accordance with your comments. I would appreciate it if you would check the revised text:
“Although this is beyond the scope of the present review, factors such as gender, ethnicity, and regionality, apart from age, may interact with SES [66-68]; thus, it is necessary to incorporate these factors into the analysis as well.”
Comment: 2) The methods section lacks some transparency. The literature source is only PubMed; why were more excluded sources excluded?
Reply: I used PubMed for the literature search because it is a medical database that aligns perfectly with the scope of this review and because its rigorous, multi-layered inclusion criteria help ensure the quality of the articles it indexes. Nevertheless, as you rightly pointed out, a systematic review should search multiple databases to achieve comprehensive coverage. I stated this limitation as follows: “Second, the purpose of this narrative review was not to provide a comprehensive overview of all studies examining the relationship between SES and health behaviors in older adults, and databases other than PubMed were not searched. To capture a broader range of relevant studies, future reviews should incorporate additional databases such as Embase, Scopus, or AMED.”
Comment: 3) There is no information as to how many articles were included/excluded.
Reply: The Methods section has been revised accordingly.
Comment: 4) In the sleep section, the limitations of COVID-era data are acknowledged (line 537), but should be emphasized more in relation to generalizability. Perhaps contrast with pre-pandemic findings if available.
Reply: According to your comment, I have added the following discussion: “It is also important to note that SES is influenced by external factors such as historical period and economic conditions. This review introduced studies conducted during the COVID-19 pandemic that examined the relationship between SES and sleep among older adults. Are there any differences compared to findings from before the pandemic? According to a review summarizing observational studies conducted between 1990 and 2019, lower SES, such as lower income or education, was associated with poorer sleep continuity and quality among older adults, consistent with findings from studies conducted during the pandemic [49]. On the other hand, a study reported that during the pandemic, approximately half of older adults experienced deteriorated sleep quality, 40% had short sleep duration, and 20% reported insomnia symptoms [63], suggesting that the relationship between SES and sleep may have been exaggerated during the pandemic. Nevertheless, the association between SES and sleep has shown consistency, and there is no doubt that addressing sleep problems among older adults with low SES can contribute to healthy aging.”
Comment: 5) In the discussion, the mention of subjective SES is compelling, but underdeveloped. You could expand on how subjective SES might interact with objective SES or other psychosocial factors in influencing health behaviors.
Reply: According to your comment, I have revised the discussion regarding subjective SES as follows:
“Subjective SES has been shown to exhibit a unique cumulative association with physical health in adults, over and above traditional objective indicators of SES [73,74]. Furthermore, there are reports that subjective SES, particularly subjective evaluations of educational attainment and occupational status, is more strongly associated with well-being than conventional SES measures, and serves as a significant predictor of psychological well-being [75]. The relationship between subjective SES and health behaviors in older adults remains insufficiently studied. Although subjective SES is beyond the scope of this review, future research should investigate the relationship between the level of subjective SES and health behaviors among older adults.”
Comment: 6) While the need to “enhance social support” is mentioned, concrete recommendations are sparse.
Reply: Several studies on social support are introduced in the Discussion section. These include financial assistance such as pensions, support in accessing healthcare services, and the provision of health education opportunities.
The relevant Discussion section has been revised as follows.
“These studies suggest that effecting meaningful changes in health behaviors and improving clinical outcomes among older adults with low SES requires large-scale, comprehensive social support including financial assistance, support for accessing healthcare services, and opportunities for health education. However, in aging societies, rising social security expenditures may place considerable strain on public finances, potentially limiting the feasibility of providing such extensive support.”
Comment: 7) It might not be measured in the studies or stated but it might be key to mention in the discussion that factors like gender, ethnicity, or rurality often interact with SES. You could briefly acknowledge that SES rarely operates in isolation and intersects with other axes of disadvantage.
I hope this feedback is helpful.
Reply: I have added the following text to the Discussion section:
Although this is beyond the scope of the present review, factors such as gender, ethnicity, and regionality, apart from age, may interact with SES [66-68]; thus, it is necessary to incorporate these factors into the analysis as well.
Thank you again for your thoughtful review.
Reviewer 3 Report
Comments and Suggestions for Authors
It is my pleasure to review the manuscript entitled “The Relationship Between Socioeconomic Status and Health Behaviors in Older Adults.” In this article, the author conducted a narrative review synthesizing evidence on how socioeconomic status (SES) influences a range of health behaviors—such as healthcare utilization, digital health engagement, use of complementary and alternative medicine (CAM), dietary patterns, physical activity, and sleep—among older adults, primarily in Japan and internationally. The paper addresses a timely and critical public health issue, especially relevant in the context of rapidly aging societies like Japan. Understanding how SES impacts health behaviors can inform policies to reduce health disparities and promote equitable aging.
- Title: The title could have been slightly enhanced by indicating that it is a narrative review, to better set reader expectations.
- Abstract: The abstract succinctly summarizes the background, methods, principal findings, and conclusions. It clearly outlines the key associations identified between higher SES and more favorable health behaviors and suggests policy interventions. However, the abstract lacks explicit mention of limitations and does not state the number of studies included or the methodological rigour of included literature, which is important for assessing the comprehensiveness of a narrative review.
- Introduction: the introduction robustly sets the stage by describing Japan as a super-aged society facing SES-related health inequities. It uses local epidemiological studies to justify the significance of exploring SES-health behavior links in older adults. The introduction frames the problem effectively by integrating specific Japanese cohort studies, which strengthens contextual relevance. The author might have expanded on gaps in existing international reviews or systematic reviews to more clearly justify the need for this specific narrative synthesis.
- Methodology: the methodology section describes a structured narrative review approach guided by Turnbull et al. It outlines a six-step framework: defining the question, justification, literature sources, search parameters, data cleaning, and synthesis. The decision to use only PubMed may limit comprehensiveness. The methodology acknowledges this as a limitation but does not mitigate it by using additional databases like Embase or Scopus. This review was conducted by a single author, increasing risk of selection and interpretive bias. As a narrative review, it inherently lacks formal quality appraisal (as would be found in a systematic review with risk-of-bias tools).
- Results: about Healthcare utilization; the review provides rich evidence on how higher SES predicts greater use of preventive services, outpatient care, and more appropriate polypharmacy. The inclusion of diverse global examples (China, Belgium, Japan, the Netherlands, USA) gives breadth. However, the synthesis could have benefited from a table explicitly detailing study designs, sample sizes, and adjustment covariates, to aid the reader in evaluating methodological heterogeneity.
About Digital health; the manuscript presents compelling evidence on the persistence of the digital divide in older adults, noting SES-linked disparities in telehealth and eHealth use. The author insightfully connects these findings to broader implications for future AI-driven healthcare, emphasizing the need for digital inclusion. However, some studies combined broad adult samples with only partial reporting for older subgroups (as noted by Miyawaki et al. and Chang et al.), slightly weakening age-specific conclusions.
About CAM and supplements; the section thoroughly covers the higher propensity for CAM and supplement use among wealthier, more educated older adults. Cross-cultural variations (e.g., Asians in the US using more CAM) are well-highlighted. It distinguishes CAM from conventional care by discussing its self-financed, belief-driven uptake, underlining why SES so strongly influences it. However, the evidence from Japan is somewhat limited, despite the primary context being Japanese older adults.
About Lifestyle (diet, PA, sleep); the paper presents a nuanced view, noting that while higher SES generally predicts healthier diets and better sleep, relationships with physical activity are mixed. For instance, some data suggest lower SES groups report higher PA post-retirement. The discussion of conflicting findings in PA by SES is an intellectually honest reflection of the literature.
- Discussion: the discussion synthesizes evidence well, arguing for policies that integrate redistributive social support with tailored digital solutions to address SES disparities. It astutely suggests that improving SES rapidly is unrealistic and instead recommends systemic interventions. By discussing specific intervention studies (e.g., transportation subsidies, copayment eliminations, telehealth in Brazil), it moves beyond documenting associations to exploring solutions. However, the review acknowledges not including subjective SES measures, which may be highly relevant for older adults whose financial income is less dynamic post-retirement.
- Limitations section
The manuscript candidly discusses its limitations: Single-author bias, Sole reliance on PubMed, and Focus on select health behaviors without covering substance use or infection control.
Author Response
Dear Reviewer 3,
Comment: It is my pleasure to review the manuscript entitled “The Relationship Between Socioeconomic Status and Health Behaviors in Older Adults.” In this article, the author conducted a narrative review synthesizing evidence on how socioeconomic status (SES) influences a range of health behaviors—such as healthcare utilization, digital health engagement, use of complementary and alternative medicine (CAM), dietary patterns, physical activity, and sleep—among older adults, primarily in Japan and internationally. The paper addresses a timely and critical public health issue, especially relevant in the context of rapidly aging societies like Japan. Understanding how SES impacts health behaviors can inform policies to reduce health disparities and promote equitable aging.
Reply: Thank you very much for taking the time to review my manuscript.
Comment: Title: The title could have been slightly enhanced by indicating that it is a narrative review, to better set reader expectations.
Reply: In accordance with your suggestion, the title was changed to “The Relationship Between Socioeconomic Status and Health Behaviors in Older Adults: A Narrative Review.”
Comment: Abstract: The abstract succinctly summarizes the background, methods, principal findings, and conclusions. It clearly outlines the key associations identified between higher SES and more favorable health behaviors and suggests policy interventions. However, the abstract lacks explicit mention of limitations and does not state the number of studies included or the methodological rigour of included literature, which is important for assessing the comprehensiveness of a narrative review.
Reply: Thank you for your comments. I have revised the Abstract and added the number of included studies; however, I was unable to mention the study limitations and methodological rigor due to the word count limit. I would appreciate it if you could review these aspects in the main text.
Comment: Introduction: the introduction robustly sets the stage by describing Japan as a super-aged society facing SES-related health inequities. It uses local epidemiological studies to justify the significance of exploring SES-health behavior links in older adults. The introduction frames the problem effectively by integrating specific Japanese cohort studies, which strengthens contextual relevance. The author might have expanded on gaps in existing international reviews or systematic reviews to more clearly justify the need for this specific narrative synthesis.
Reply: According to your suggestion, I have added the following text to the Introduction section: “Outside Japan, systematic reviews have also reported that low SES is significantly associated with poorer outcomes in emergency care [11], excessive gestational weight gain [12], and higher incidence of frailty [13]. However, there are very few reviews that specifically examine the relationship between SES and health behaviors among older adults.”
Comment: Methodology: the methodology section describes a structured narrative review approach guided by Turnbull et al. It outlines a six-step framework: defining the question, justification, literature sources, search parameters, data cleaning, and synthesis. The decision to use only PubMed may limit comprehensiveness. The methodology acknowledges this as a limitation but does not mitigate it by using additional databases like Embase or Scopus. This review was conducted by a single author, increasing risk of selection and interpretive bias. As a narrative review, it inherently lacks formal quality appraisal (as would be found in a systematic review with risk-of-bias tools).
Reply: I completely agree with you. This point is discussed in detail under the study limitations.
Comment: Results: about Healthcare utilization; the review provides rich evidence on how higher SES predicts greater use of preventive services, outpatient care, and more appropriate polypharmacy. The inclusion of diverse global examples (China, Belgium, Japan, the Netherlands, USA) gives breadth. However, the synthesis could have benefited from a table explicitly detailing study designs, sample sizes, and adjustment covariates, to aid the reader in evaluating methodological heterogeneity.
Reply: I would appreciate it if you would check the summary tables (Table 1-4).
Comment: About Digital health; the manuscript presents compelling evidence on the persistence of the digital divide in older adults, noting SES-linked disparities in telehealth and eHealth use. The author insightfully connects these findings to broader implications for future AI-driven healthcare, emphasizing the need for digital inclusion. However, some studies combined broad adult samples with only partial reporting for older subgroups (as noted by Miyawaki et al. and Chang et al.), slightly weakening age-specific conclusions.
Reply: Thank you for your careful reading. To facilitate reader understanding, the proportion of older adult participants has been noted for each study.
Comment: About CAM and supplements; the section thoroughly covers the higher propensity for CAM and supplement use among wealthier, more educated older adults. Cross-cultural variations (e.g., Asians in the US using more CAM) are well-highlighted. It distinguishes CAM from conventional care by discussing its self-financed, belief-driven uptake, underlining why SES so strongly influences it. However, the evidence from Japan is somewhat limited, despite the primary context being Japanese older adults.
Reply: The low number of CAM users in Japan may be one of the reasons why appropriate studies could not be identified in this review. In Japan, the proportion of CAM users is 12.8%, which appears to be quite low from an international perspective [JMA J. 2019;2(1):35-46.].
Comment: About Lifestyle (diet, PA, sleep); the paper presents a nuanced view, noting that while higher SES generally predicts healthier diets and better sleep, relationships with physical activity are mixed. For instance, some data suggest lower SES groups report higher PA post-retirement. The discussion of conflicting findings in PA by SES is an intellectually honest reflection of the literature.
Reply: I appreciate your kind evaluation.
Comment: Discussion: the discussion synthesizes evidence well, arguing for policies that integrate redistributive social support with tailored digital solutions to address SES disparities. It astutely suggests that improving SES rapidly is unrealistic and instead recommends systemic interventions. By discussing specific intervention studies (e.g., transportation subsidies, copayment eliminations, telehealth in Brazil), it moves beyond documenting associations to exploring solutions. However, the review acknowledges not including subjective SES measures, which may be highly relevant for older adults whose financial income is less dynamic post-retirement.
Reply: I agree with you. Research on the relationship between subjective SES and objective SES appears to be insufficient. In particular, when examining the association between SES and health behaviors among older adults, it is essential to investigate subjective SES and self-rated health.
Limitations section
Comment: The manuscript candidly discusses its limitations: Single-author bias, Sole reliance on PubMed, and Focus on select health behaviors without covering substance use or infection control.
Reply: I hope that evidence regarding the SES of older adults will continue to accumulate in the future, leading to findings with a higher level of evidence. This will enable people to lead sustainable, healthy, and fulfilling lives in an increasingly super-aged society.
Round 2
Reviewer 1 Report
Comments and Suggestions for Authors
All of concerns regarding this study have been solved.
Good luck for the future submission.
Reviewer 2 Report
Comments and Suggestions for Authors
Thank you for revising the manuscript and good luck with the rest of the process.